# Outcomes Following Autologous Fat Grafting in Patients with Sequelae of Head and Neck Cancer Treatment

**DOI:** 10.3390/cancers15030800

**Published:** 2023-01-28

**Authors:** Jorge Masià-Gridilla, Javier Gutiérrez-Santamaría, Iago Álvarez-Sáez, Jorge Pamias-Romero, Manel Saez-Barba, Coro Bescós-Atin

**Affiliations:** 1Noves Tecnologies i Microcirurgia Craniofacial, Vall d’Hebron Institut de Recerca (VHIR), Hospital Universitari Vall d’Hebron, Vall d’Hebron Barcelona Hospital Campus, Passeig Vall d’Hebron 119-129, E-08035 Barcelona, Spain; 2Servei de Cirurgia Oral i Maxil·lofacial, Hospital Universitari Vall d’Hebron, Vall d’Hebron Barcelona Hospital Campus, Passeig Vall d’Hebron 119-129, E-08035 Barcelona, Spain; 3Unitat Docent Vall d’Hebron, Facultat de Medicina, Universitat Autònoma de Barcelona, Passeig Vall d’Hebron 119-129, E-08035 Barcelona, Spain

**Keywords:** autologous fat grafting, head and neck cancer, radiotherapy, reconstruction, sequelae, quality of life

## Abstract

**Simple Summary:**

In recent years, there have been relevant advances in the use of surgery, radiation therapy, and chemotherapy for the treatment of malignant tumors of the head and neck. Extensive tumor resection and radical radiotherapy frequently result in altered form and function of orofacial structures that can severely impact the patient’s quality of life. This study reports the benefits obtained with the injection of autologous fat to correct the deformities and improve functionality in a series of 40 patients who have been treated for head and neck cancer. Esthetic improvement was obtained in 77.5% of patients and functional improvement in 89.2%. In addition, there was a high degree of satisfaction regarding esthetic improvement and 92.5% of patients would recommend the procedure to other patients in the same situation. The injection of autologous fat is an effective procedure for the management of sequelae of head and neck cancer treatment.

**Abstract:**

A single-center retrospective study was designed to assess the outcomes of autologous fat grafting for improving surgery- and radiotherapy-related sequelae in 40 patients with head and neck cancer. All patients underwent surgical resection of primary tumors and radiotherapy (50–70 Gy) and were followed over 12 months after fat grafting. Eligibility for fat grafting procedures included complete remission after at least 3 years of oncological treatment. The cervical and paramandibular regions were the most frequently treated areas. Injected fat volumes ranged between 7.5 and 120 mL (mean: 23 mL). Esthetic improvement was obtained in 77.5% of patients, being significant in 17.5%, and functional improvement in 89.2%, being significant in 29.7% of patients. Minor complications occurred in three patients. There was a high degree of satisfaction regarding esthetic improvement, global satisfaction, and 92.5% of patients would recommend the procedure. This study confirms the benefits of fat grafting as a volumetric correction reconstructive strategy with successful cosmetic and functional outcomes in patients suffering from sequelae after head and neck cancer treatment.

## 1. Introduction

Head and neck cancer represents the seventh most common cancer worldwide, with 1.1 million new diagnoses reported annually [1,2]. However, there is a substantial geographical variation in the incidence and anatomical distribution of tumors, predominantly attributed to differences in smoking and alcohol consumption, steady increase in human papillomavirus-related cancer, genetic predisposition, or exposure to ionizing radiation, which are known to play an important role in the pathogenesis of the disease [3,4]. Radiation therapy, surgery or both combined and chemotherapy are currently available standard therapeutic strategies but are often prioritized differently depending on the site of tumor origin, histological diagnosis, tumor burden, quality of life considerations, patient preference, or hospital characteristics with the availability of specialized multidisciplinary care teams [5,6].

Advances in surgery, radiation therapy, and chemoradiotherapy have improved locoregional control and survival, but the outcomes of these treatment modalities have incorporated preservation and restoration of function in the focus of radical ablation and curative efforts [7]. However, despite improvements in the multimodal treatment approach aimed at decreasing cosmetic and functional deficits with resultant psychological, physical, and nutritional detriments [8,9], management of sequelae following treatment of head and neck cancer, particularly in patients with locally advanced tumors, still remains a challenge difficult to solve in daily practice [10,11].

Fat grafting, also referred to as fat transfer or fat injections, dates back to 1893 when Neuber first described the technique and reported successful outcomes after transplanting fat beneath atrophic scars [12]. Structural autologous fat grafts for the enhancement of facial contours were proposed by Coleman in 1997 [13] and the Coleman’s lipostructure technique became subsequently recognized as a standard procedure for fat transfer [14,15]. In recent years, autologous fat grafting has been described by different authors as a very useful tool to improve residual esthetic and functional deformities after head and neck cancer treatment, and for its ability to correct volumetric defects and regenerative properties [16,17,18,19]. A systematic review and meta-analysis of 52 studies with 1568 patients confirmed that autologous fat transfer is an effective technique in facial reconstruction surgery with a low rate of minor complications [20].

The purpose of this study was to evaluate esthetic and functional outcomes as well as patients’ satisfaction and complications associated with autologous fat grafting in the context of integral management of head and neck cancer patients.

## 2. Materials and Methods

### 2.1. Design and Study Population

A retrospective study was made of all consecutive patients who required autologous fat grafting procedures between January 2010 and January 2019 at the Service of Oral and Maxillofacial Surgery of Hospital Universitari Vall d’Hebron, in Barcelona, Spain. Fat grafting was indicated for the treatment of sequelae associated with any form of therapy of head and neck cancer. Inclusion criteria were history of head and neck cancer treated with surgery, radiation therapy, or chemotherapy followed by duration of complete clinical remission of at least 3 years; presence of severe or very severe esthetic defects and/or loss of skin flexibility, and severe or very severe dysphonia, dysphagia, alteration in head and neck mobility, and alteration in swallowing or chewing, corresponding to scores 3 or 4 of esthetic and/or functional evaluation of the scoring method described by Pulphin et al. [21]; good health, as confirmed by preoperative work-up studies; and signed informed consent. Patients previously treated with fat infiltration procedures or with insufficient fat tissue deposits for fat transfer were excluded from the study, as were those expected to have poor adherence to follow-up visits scheduled for at least 12 months after the intervention, and ineligibility as judged by the investigators.

The study protocol was approved by the Clinical Research Ethics Committee of Hospital Universitari Vall d’Hebron (code PR (ATR) 57/2016, approval date 26 February 2016) (Barcelona, Spain). Written informed consent was obtained from all participants.

### 2.2. Fat Grafting and Surgical Procedure

The available fat deposits were evaluated, and the donor site was selected with consent from the patient. Fat harvesting was performed under general anesthesia or local anesthesia with intravenous sedation, and the patient was in the supine position. Ten minutes before liposuction, abdominal infiltration was performed through a 2–3 mm incision puncture at the level of both flanks or in the umbilical region, using a modified Klein solution (500 mL Ringer lactate) with 0.5 mg epinephrine, and adding 1% lidocaine for patients under sedation. Harvesting was performed through the same infiltration incisions using a liposuction cannula (COL-ASP15, 3 mm × 15 cm, Byron Medical Inc., Tucson, AZ, USA or COL-KHU12 Mitmed^®^, 3 mm × 20 cm, Surgest Medical, Sant Cugat del Vallès, Barcelona, Spain) connected to a 10 mL Luer-Loc syringe, by firm and regular back-and-forth movements under low negative digital pressure until the syringe was filled. Then, fat was purified either by the centrifugation method described by Coleman [22] (3000 rpm for 3 min) (Medigraft-BL^®^ Centrifuge, Surgest Medical) or washing and filtration using the Puregraft system (Cytori Therapeutics, San Diego, CA, USA).

The graft was injected in small amounts, separated between them in order to obtain a better vascularization and therefore longer graft survival, slowly and without overcorrection, from the deep to the superficial cutaneous tissue using an atraumatic cannula (7–9 mm long, 16G, types I-III COL-19, Byron Medical, COL SPA9), creating multiple tunnels in a fan-like fashion following the recommendation of Coleman [22]. An abdominal bandage was applied for 48 h and substituted with an abdominal belt for the following 7 days. Amoxicillin–clavulanic acid (500/250 mg), 1 tablet every 8 h, was administered during the first 7 days after the procedure.

### 2.3. Evaluation and Follow-Up

Patients were visited postoperatively by the same investigator (J.M.-G.) after 1 week of fat grafting and at 1, 3, 6, and 12 months thereafter. At each visit, patients were questioned and underwent a physical examination to assess the evolution of the graft and the eventual appearance of early or late complications. Twelve months after fat grafting, esthetic and functional results were evaluated using a 4-point scale described by Pulphin et al. [21], including no esthetic or functional problems (score 0), and esthetic defects and/or loss of skin flexibility and functional alterations of dysphonia, dysphagia, neck/head mobility or swallowing or chewing scored as 1 = slight, 2 = moderate, 3 = severe, and 4 = very severe. Improvement was defined in the presence of a postoperative score lower than the preoperative score, and significant improvement was defined if the postoperative score was 2 or more points lower than the preoperative score. The severity of complications was classified according to the Clavien–Dindo classification system [23]. 

Also, after 12 months of fat grafting, the patient’s satisfaction regarding esthetic improvement was evaluated on a scale of 0 to 10 (0 = nothing, 10 = maximum satisfaction) and the overall satisfaction with treatment according to responses to the following four questions: “*What is the degree of satisfaction with the treatment received?*”, “*Do you consider that you have received sufficient and clear information?*”, “*Did the treatment meet your expectations?*”, and “*In case you request an advice, would you recommend this treatment to another patient in the same conditions?*” using a 4-point Likert scale (1 = nothing, 2 = little, 3 = quite a lot, 4 = a lot/very much).

### 2.4. Statistical Analysis

Descriptive statistics were calculated and presented as frequencies and percentages for categorical variables, and as mean and standard deviation (SD) for continuous variables.

## 3. Results

The study population included 40 patients, 26 men and 14 women, with a mean age of 60.5 years (range 32–86 years). Complete data at the 12-month follow-up visit were obtained in all participants. Squamous cell carcinoma of the oral cavity was the most frequent primary tumor (*n* = 25, 62.5%) followed by a malignant tumor of the salivary glands (*n* = 7, 17.5%). All patients underwent radical surgery of the neoplasms and radiotherapy (50–70 Gy), and 23 of them (57.5%) received chemotherapy. Reconstruction procedures using different types of flaps were performed in 27 (67.5%) patients using microsurgical free fibula flaps in most of them. All patients presented with esthetic sequelae, including scarring, cervicofacial asymmetry, and cutaneous fibrosis. Limitations of neck movements, trismus, and dysphagia were the most frequent functional sequelae.

The abdominal region was the donor site for fat grafting in all patients. The manual low pressure aspiration technique was used to obtain the fat graft in all cases using a 10 mL syringe with Luer-Loc connection and COL-KHU12 Mitmed^®^, 3 mm × 20 cm liposuction cannula. Regarding the processing method, the centrifugation method following Coleman’s recommendations [22] was used in the first series of 16 patients, and the filtration processing system was carried out using the Puregraft device in the remaining 24. In all patients, infiltration was performed following the Coleman technique [22].

Fat grafting was mostly performed under general anesthesia, with sedation and local anesthesia in only three patients. The cervical and paramandibular regions were the most frequently treated areas, with injected fat volumes ranging between 7.5 and 120 mL (mean: 23 mL). The length of surgery varied between 45 and 180 min, with a mean of 89 min. No intraoperative complications were recorded, and all patients were discharged within 24 h after the procedure. Minor complications occurred in three patients (7.5%) with abdominal pain, seroma, and lingual paresthesia in one patient each, and they resolved spontaneously. All these complications were classified as grade I of the Clavien–Dindo classification system [23].

Esthetic improvement was obtained in 31 patients (77.5%), being significant in 7 of them (17.5%). In relation to functional alterations, there were three patients who scored 0 preoperatively. In the remaining 37 patients, functional improvement was found in 33 (89.2%), being significant in 11 of them (29.7%). One of the most widespread findings in the treated patients was clinical improvement in the quality of irradiated skin on the neck or face, with apparent improvement in blood supply, skin smoothness, function, and elasticity. The analysis of graft stability was performed clinically by evaluating the patient and analyzing the photographic documentation, showing a progressive volumetric decrease close to 50% of the injected volume in all patients. Details of treatment characteristics and outcome of the study patients are shown in Table 1. Figure 1, Figure 2, Figure 3, Figure 4 and Figure 5 show some representative cases. Postoperative photographs of these patients were obtained between 6 and 12 months of follow-up after the fat grafting procedure.

Esthetic improvement evaluated by the patients showed a mean (SD) score of 7.03 (1.83) and a mean satisfaction with treatment of 3.05 (0.68). In addition, 37 patients (92.5%) would recommend treatment with autologous fat grafting to other patients in a similar situation.

After 12 months of follow-up of autologous fat grafting, two patients died; the causes of death were a new lung cancer and heart disease, respectively. Recurrence of the primary head and neck cancer occurred in three patients, but in all cases, the site of recurrence was far from the fat infiltrated area.

## 4. Discussion

Autologous fat grafting is a feasible and valuable technique for patients with sequelae following surgery and radiotherapy of primary head and neck cancer tumors [17,24]. However, the experience with the use of fat grafting for esthetic and functional improvement in these patients is still limited [17,18,19,20,21,22,25]. The present clinical series is the largest published of head and neck cancer patients treated with combined surgery and radiation therapy of at least 50 Gy, undergoing autologous fat grafting for the correction of esthetic and functional sequelae of treatment. In all cases, fat grafting was performed after a disease-free interval of 3 years, a time period with the highest risk of tumor recurrence. In other studies, fat grafting has been performed after a minimum follow-up of 1 year [19,26].

All patients were operated on following the technique described by Coleman [13,22], although in 60% of cases (*n* = 14), purification was performed using the Puregraft system as it was considered that this method better preserved the sterility of the circuit and eliminated the exposure of fat to air, thus avoiding rapid desiccation and preserving the survival of adipocytes. Zhu et al. [27] have compared three preparation methods for fat grafts in twenty-two donors: gravity separation, Coleman centrifugation, and simultaneous washing with filtration using the Puregraft system. Grafts prepared by washing with filtration exhibited significantly reduced blood cell and free lipid content, with significantly greater adipose tissue viability than other methods.

In our study, esthetic and functional outcomes were evaluated at 12 months after fat grafting using the 4-point scale described by Pulphin et al. [21]. Esthetic improvement was obtained in 77.5% of patients, being significant in 17.5%, and functional improvement in 89.2%, being significant in 29.7% of patients (significant improvement defined as postoperative score of 2 or more points lower than preoperative score). The rate of improvement is difficult to compare to other previously published studies because of differences in the scoring system for the assessment of results, except for similar findings in a preliminary feasibility study of 12 patients reported by our group [17], and the clinical series of 11 patients reported by Pulphin et al. [21] who were the authors that described the evaluation score system. In this study, significant esthetic improvement was obtained in nine patients (81.8%) and functional improvement in seven (63.6%). The total injected volume ranged between 10 and 119 mL, with an average of 48.5 mL, which is a somewhat greater volume than the 23 mL used in our study. No complications were recorded. Patients were followed for a mean of 39.9 months (range 2–88 months), but the resorption of engrafted fat was observed for all patients and was estimated to be approximately 20% to 40%. Because of the importance of the defects, reinjection was performed in six patients. In addition, histological examination of biopsies taken from the treated areas of six patients showed reduction in irradiated morphology patterns, with normal histological structure, high vascular network density, and reduction in fibrosis. In our study, biopsies from fat grafting areas were not obtained.

In 2003, Ducic et al. [19] reported data of a retrospective series of 23 patients undergoing lipotransfer as part of their craniofacial reconstructive procedure. In this study, six patients underwent a total of eight fat transfer procedures (two procedures in two patients), with good results in five and inadequate results in one. No intraoperative or postoperative complications were observed. Vitagliano et al. [26] described 10 patients with squamous and basal cell carcinomas of the lower or upper lips treated with resection and nasolabial flaps. After 6 months of the primary surgery, 5 of these 10 patients underwent fat grafting to improve persistent depressions and deformities. All treated patients showed favorable cosmetic and favorable results in terms of improvement of their clinical appearance, oral competence, sensitivity, and lip movements. In the study of Karmali et al. [18], 116 patients with head and neck cancer (or benign locally aggressive tumors), with history of radiotherapy in 69% of cases, underwent 190 fat grafting procedures. However, the esthetic outcomes were evaluated in only 17 patients after a mean follow-up of over 2 years, with significant improvements in all of them according to a 5-point Likert scale as evaluated by 10 plastic surgeons and 10 laypersons. Procedural-related complications were observed in 5.1% of cases (infection, oil cysts, fat necrosis) and all four locoregional recurrences were in areas outside of where the fat was grafted. Griffin et al. [28] reported a retrospective analysis of 38 patients who underwent fat grafting, with a history of head and neck malignancy, multimodal treatment including at least surgery or radiotherapy, and at least 2-year disease-free survival. Esthetic and functional improvements in their radiation-induced skin fibrosis, and volumetric defects at a follow-up of 32 months were shown in 37 (97%) patients. Lipotransfer was also associated with psychological and quality of life improvement. In this study, recurrence was detected in two patients (5.3%) after a mean follow-up of 10 years.

Patients’ satisfaction was also evaluated in our study, showing a high degree of satisfaction in terms of esthetic improvement, global satisfaction with treatment, and percentage of patients who would recommend fat grafting to other patients in similar conditions.

However, despite the refinement of technical aspects of lipotransfer and encouraging results for improving esthetic and functional sequelae of surgery and radiotherapy in head and neck cancer patients, the variability of fat absorption rates has been recognized as a limitation of the procedure. Although the restoration of altered contour can be achieved reproducibly intraoperatively and in the early postoperative period, the long-term durability of results remains to be established. Moreover, methods for quantifying the stability of grafted fat have not been standardized. Hörl et al. [29] reported an average volume decline of 55% at 6 months, evaluated by resonance magnetic imaging (RMI) studies in a group of 53 patients with facial defects repaired using autogenous fat tissue. Meier et al. [30] provided three-dimensional volumetric measurements demonstrating an average graft survival of 32% at 16 months after autologous fat grafting for midfacial rejuvenation. However, Coleman [25] indicates that the volume of the graft stabilizes at 3–4 months, and a subtle volumetric decrease may occur up to 1 year after infiltration; beyond that, he states that the volume remains constant for 8–12 years. Quantifiable data of graft survival are rarely reported. In our series, clinical examination and comparison of photographs over the follow-up period showed a progressive volumetric decrease, close to 50% of the injected volume.

Although recurrences in our study, like others [18,28], occurred in areas far from the treatment site, the use of autologous fat grafting in a bed with a history of cancer involvement is a matter of concern. Further clinical studies with longer follow-up periods are needed to confirm these findings. Finally, patients should be informed regarding the possibility of having to repeat fat grafting in order to achieve more stable and visible results.

## 5. Conclusions

Autologous fat grafting is a valuable technique for improving esthetic and functional sequelae of extensive surgical resections and radiation therapy in patients with malignant head and neck tumors. The technique is a minimally invasive procedure for which a sufficient volume of abdominal fat can be easily obtained. The results of the present study confirm the benefits of fat grafting as a volumetric correction reconstructive strategy, with successful cosmetic and functional outcomes, a high degree of patient satisfaction, low complication rate, and no evidence of being associated with cancer recurrence. 

## Figures and Tables

**Figure 1 cancers-15-00800-f001:**
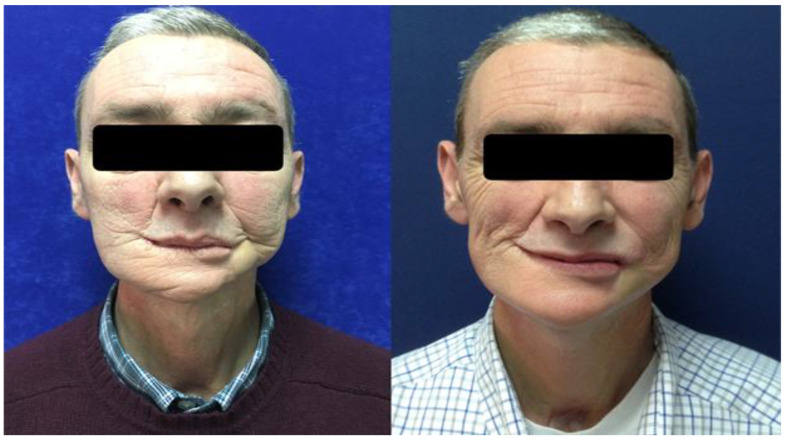
A 59-year-old male patient treated with left segmental mandibulectomy and microsurgical reconstruction with a microsurgical fibula flap, ipsilateral cervical lymph node dissection, and postoperative radiotherapy at a dose of 66 Gy for a squamous cell carcinoma of the left alveolar crest. He was treated with autologous fat grafting in the paramandibular and cervical regions with a total of 24 mL of fat (**left**). The postoperative photograph at follow-up shows the improvement of the paramandibular and cervical deformity (**right**).

**Figure 2 cancers-15-00800-f002:**
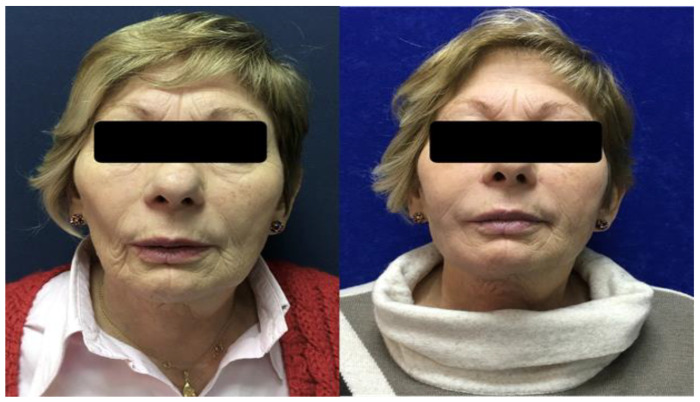
A 71-year-old female patient treated with right buccal mucosa excision, and postoperative radiotherapy at a dose of 66 Gy for an oral squamous cell carcinoma. She was treated with autologous fat grafting in cheek and cervical regions with a total of 23 mL of fat (**left**). The postoperative photograph at follow-up shows the improvement of the facial and cervical deformity (**right**).

**Figure 3 cancers-15-00800-f003:**
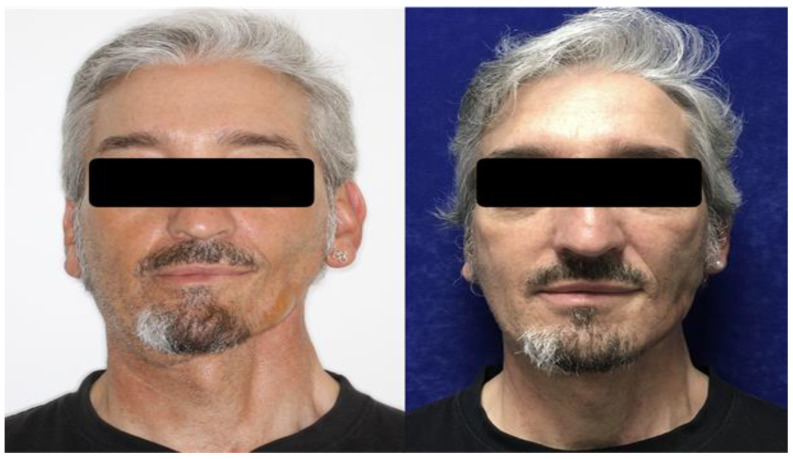
A 54-year-old male patient treated for left segmental mandibulectomy and microsurgical reconstruction with a microsurgical fibula flap, ipsilateral cervical lymph node dissection, and postoperative radiotherapy at a dose of 64 Gy for a squamous cell carcinoma of the left alveolar crest. He was treated with autologous fat grafting in the paramandibular and cervical regions with a total of 23 mL of fat (**left**). The postoperative photograph at follow-up shows the improvement of the paramandibular and cervical deformity (**right**).

**Figure 4 cancers-15-00800-f004:**
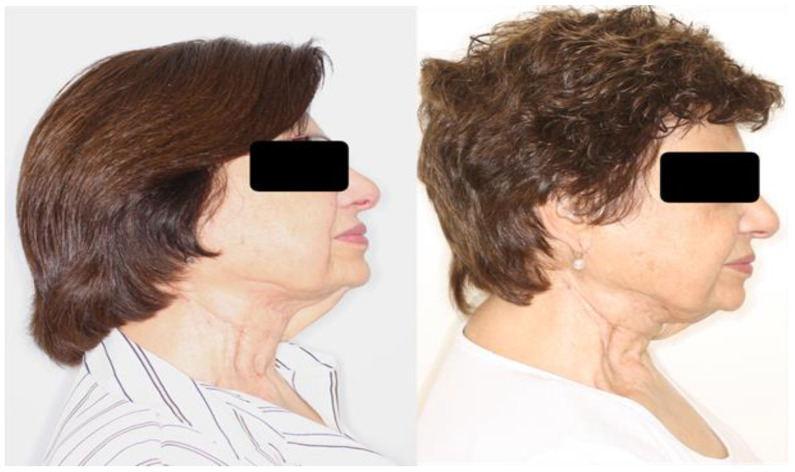
A 71-year-old female patient treated for right buccal squamous cell carcinoma with local resection and ipsilateral radical cervical lymph node dissection and postoperative radiotherapy at a dose of 66 Gy for a squamous cell carcinoma. She was treated with autologous fat grafting in the right cervical region with a total of 23 mL of fat (**left**). The postoperative photograph at follow-up shows the improvement of the paramandibular and cervical deformity (**right**).

**Figure 5 cancers-15-00800-f005:**
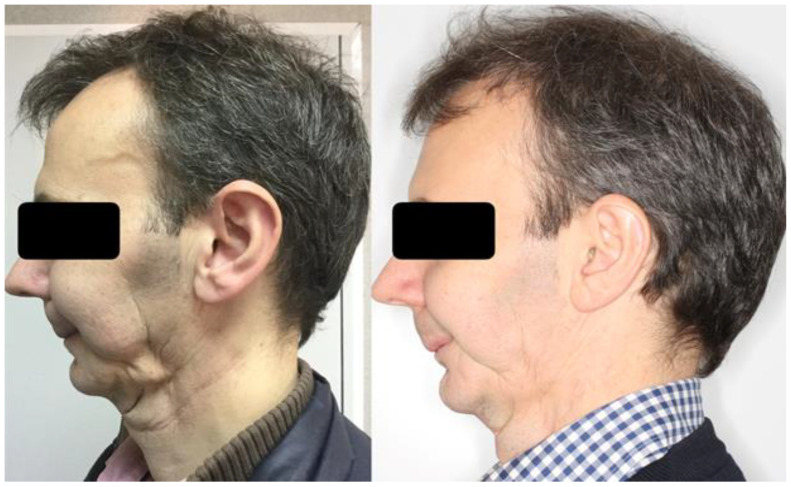
A 43-year-old male patient treated for an intraosseous carcinoma of the left mandible with segmental mandibulectomy and microsurgical reconstruction with a fibula flap, ipsilateral cervical lymph node dissection, and postoperative radiotherapy at a dose of 63 Gy. He was treated with autologous fat grafting in the paramandibular and cervical regions with a total of 24 mL of fat (**left**). The postoperative photograph at follow-up shows the improvement of the paramandibular and cervical deformity (**right**).

**Table 1 cancers-15-00800-t001:** Clinical characteristics, details of treatment, and outcome in the 40 study patients.

Patient	Histology /Location	Surgery	Chemotherapy	RT Gy	Reconstruction Type	Injection Site	Volume mL	Length of Surgery min	Anesthesia	Esthetic Score Preoperative/ Postoperative	Functional Score Preoperative/ Postoperative
1	ACC/parotid	PT + MD	No	60	Not performed	Laterocervical and parotid area	7.5	50	General	4/3	3/2
2	DFSP/malar	TEOM	No	60	Mustarde cheek flap	Hemifacial	23	45	General	3/2	3/2
3	SCC/tongue	MD + ND	Yes	70	Microsurgical fibula flap	Laterocervical and paramandibular	28	65	General	4/4	4/3
4	SCC/gums	MD + ND	Yes	60	Not performed	Laterocervical and paramandibular	20	120	General	4/2	2/1
5	SCC/gums	MD + ND	No	66	Microsurgical fibula flap	Submaxillary, lower lip, nasolabial and submental fold	24	64	General	3/2	3/2
6	SCC/gums, mouth floor	MD + ND bilateral	Yes	70	Fibula flap + anterolateral thigh flap	Laterocervical and paramandibular	19	90	General	4/4	4/3
7	SCC/gums	MD + ND	Yes	50	Not performed	Laterocervical and paramandibular	24	115	General	4/3	3/2
8	SCC/jugal mucosa	TEOM	Yes	60	Local flap	Jugal region, nasolabial, submental. Laterocervical, tracheocervical	20	100	General	3/2	4/2
9	SCC/gums	TEOM + MD + ND	Yes	60	Microsurgical fibula flap	Laterocervical and paramandibular	37	135	General	4/4	3/2
10	SCC/gums	MD + ND	Yes	70	Fibula flap	Laterocervical and paramandibular	42	120	General	3/3	3/3
11	SCC/tongue, mouth floor	TEOM + MD	No	70	Fibula flap + anterolateral thigh flap	Hemifacial and cervical	20	80	Sedation local	4/3	3/2
12	SCC/retromolar trigone	MD + ND	No	70	Microsurgical fibula flap	Hemifacial and cervical	27	180	General	4/3	3/2
13	SCC/jugal mucosa	TEOM + ND	No	60	Radial flap	Laterocervical, paramandibular, jugal	23	110	General	3/2	2/1
14	SCC/mouth floor	MD + ND	Yes	70	Microsurgical fibula flap	Laterocervical, paramandibular, tracheal	70	100	General	4/3	4/2
15	Myoepithelial carcinoma/ minor salivary gland, maxillary	Maxillectomy	No	60	Temporal muscle flap	Temporal	13	150	General	3/1	0/0
16	Myoepithelial carcinoma/parotid gland	Radical Parotidectomy + ND	Yes	66	Not performed	Laterocervical and parotid region	8.5	58	General	3/2	2/1
17	SCC/gums	MD + ND	Yes	70	Microsurgical fibula flap	Hemifacial and cervical	15	110	General	3/2	3/1
18	SCC/gums	Maxillectomy + ND	Yes	64	Microsurgical fibula flap	Hemifacial and cervical	23	70	General	3/2	3/2
19	Adenocarcinoma parotid gland	Total parotidectomy	No	50	Not performed	Hemifacial	13	87	General	2/0	2/0
20	SCC/retromolar trigone	MD + ND	Yes	70	Not performed	Nasolabial fold, upper and lower lip, jugal and laterocervical	120	170	General	4/3	4/2
21	SCC/cervical unknown origin	ND	No	60	Not performed	Laterocervical	15	52	Sedation local	3/4	4/4
22	SCC/tongue, mouth floor	Glossectomy + MD + ND	No	66	Microsurgical fibula flap	Upper and lower lips, paramandibular, superior laterocervical, bilateral submandibular, bilateral nasolabial folds	20	117	General	4/2	3/1
23	SCC/mouth floor	MD + ND bilateral	No	70	Microsurgical fibula flap	Lower lip, paramandibular, laterocervical	12	97	General	4/2	3/2
24	Ductal carcinoma/parotid gland	Superficial parotidectomy	No	60	Not performed	Laterocervical and parotid region	15	71	General	2/1	2/1
25	SCC/retromolar trigone	MD + ND	Yes	70	Microsurgical fibula flap	Paramandibular, submaxillary, upper and lower lip	23	63	General	4/3	4/2
26	SCC/jugal mucosa	TEOM + ND	Yes	66	Local flap	Laterocervical, paramandibular, jugal	23	79	General	3/2	4/2
27	SCC/mandibular intraosseous	MD + ND bilateral	Yes	63	Microsurgical fibula flap	Paramandibular, laterocervical, nasolabial fold, lower lip	23	103	General	4/2	4/2
28	Undifferentiated parotid carcinoma	Total parotidectomy	No	60	Not performed	Paramandibular and parotid region	25	83	General	4/2	3/1
29	SCC/cervical unknown origin	ND	Yes	60	Not performed	Laterocervical	14	47	General	3/2	4/3
30	SCC/tongue	Glossectomy + ND	Yes	54	Nor performed	Laterocervical, lingual	18	115	General	3/3	4/4
31	ACC/minor salivary gland maxillary	Maxillectomy	No	66	Temporal muscle flap	Malar bilateral, left nasolabial fold, upper lip, left jugal	20	47	General	3/2	0/0
32	SCC/gums	MD + ND	Yes	70	Pectoral flap	Paramandibular and jugal	10	48	General	4/2	2/1
33	SCC/retromolar trigone	MD + ND	Yes	70	Pectoral flap + fibula flap	Hemifacial, cervical, labial, tracheal	15	114	General	4/2	3/2
34	SCC/lip	TEOM + ND	No	55	Anterolateral thigh flap	Jugal and labial	8	54	Sedation local	2/2	0/0
35	ACC/oropharynx-tongue	Glossectomy + ND	No	66	Anterolateral thigh flap	Laterocervical	20	104	General	3/3	3/3
36	SCC/gums, mouth floor	MD + ND	Yes	69	Not performed	Right laterocervical, paramandibular	15	55	General	4/4	4/2
37	Angiofibroma/nasal	TEOM	No	50	Not performed	Temporal	20	95	General	3/2	1/0
38	SCC/mandibular symphysis	MD + ND bilateral	Yes	70	Microsurgical fibula flap	Laterocervical and submental	10	48	General	4/3	3/2
39	Osteosarcoma mandibular	MD + maxillectomy + ND	Yes	60	Anterolateral thigh flap	Hemifacial and cervical	10	70	General	4/3	3/2
40	ACC/upper maxilla	Maxillectomy + ND	Yes	72	Temporal muscle flap + microsurgical fibula flap	Hemifacial, cervical and temporal	27	76	General	3/2	3/1

ACC: adenoid cystic carcinoma, DFSP: dermatofibrosarcoma protuberans, MD: mandibulectomy, ND: neck dissection, PT: parotidectomy, RT: radiotherapy, SCC: squamous cell carcinoma, TEOM: tumoral exeresis with oncological margins.

## Data Availability

Clinical data are not shared.

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
