# Peer review of "Outcomes Following Autologous Fat Grafting in Patients with Sequelae of Head and Neck Cancer Treatment"

_cancers, 2023, doi:10.3390/cancers15030800_

Round 1

Reviewer 1 Report

I think this is a nice study.

Reviewer 2 Report

This is a good article on autologous fat grafting in head and neck cancer patients. It is very difficult to evaluate scientifically the results of these procedures, so the authors should be congratuled for their efforts to provide objective measurments.  The only remark is that the delay between surgery and post operative photographs is not specified: early photographs are of course more spectacular, but consecutive photographs showing the decrease of the volume with time would have been more informative.

These procedures often need to be proposed more than once to achieve a more stable and visible result. This was not apparently proposed to the patients in this study. It could have been discussed

Reviewer 3 Report

The current study is noteworthy as minimally invasive procedures have resulted in increased use of autologous fat grafting for head and neck tumors. This is a well described case series. Strengths include the relatively large numbers, standardized follow-up (interval) times and patient-obtained satisfaction scales pre and post follow-up.  Harvesting was done using a common procedure allowing for generalizablity of findings, and esthetics and functionality were evaluated appropriately. 
